# Synovial Fluid Derived from Human Knee Osteoarthritis Increases the Viability of Human Adipose-Derived Stem Cells through Upregulation of FOSL1

**DOI:** 10.3390/cells12020330

**Published:** 2023-01-15

**Authors:** Hironori Kitajima, Takuya Sakamoto, Tetsuhiro Horie, Ayane Kuwano, Atsushi Fuku, Yasuhiko Taki, Yuka Nakamura, Ikuhiro Tanida, Hiroshi Sunami, Hiroaki Hirata, Yoshiyuki Tachi, Naoki Yamamoto, Yasuo Iida, Yasuhito Ishigaki, Sohsuke Yamada, Shigetaka Shimodaira, Yusuke Shimizu, Toru Ichiseki, Ayumi Kaneuji, Satoshi Osawa, Norio Kawahara

**Affiliations:** 1Department of Orthopedic Surgery, Kanazawa Medical University, Kahoku 920-0293, Ishikawa, Japan; 2Medical Research Institute, Kanazawa Medical University, Kahoku 920-0293, Ishikawa, Japan; 3Genome Biotechnology Laboratory, Kanazawa Institute of Technology, Hakusan 924-0838, Ishikawa, Japan; 4Faculty of Medicine, Advanced Medical Research Center, University of the Ryukyus, Nakagami 903-0215, Okinawa, Japan; 5Support Office for Bioresource Research, Research Promotion Headquarters, Fujita Health University, Toyoake 470-1192, Aichi, Japan; 6Department of Mathematics, Division of General Education, Kanazawa Medical University, Kahoku 920-0293, Ishikawa, Japan; 7Center for Regenerative Medicine, Kanazawa Medical University Hospital, Kahoku 920-0293, Ishikawa, Japan; 8Department of Pathology and Laboratory Medicine, Kanazawa Medical University, Kahoku 920-0293, Ishikawa, Japan; 9Department of Pathology, Kanazawa Medical University Hospital, Kahoku 920-0293, Ishikawa, Japan; 10Department of Regenerative Medicine, Kanazawa Medical University, Kahoku 920-0293, Ishikawa, Japan; 11Department of Plastic and Reconstructive Surgery, Graduate School of Medicine, University of the Ryukyus, Nakagami 903-0215, Okinawa, Japan

**Keywords:** synovial fluid, human knee osteoarthritis, FOSL1, human adipose-derived stem cell

## Abstract

Knee osteoarthritis (Knee OA) is an irreversible condition that causes bone deformity and degeneration of the articular cartilage that comprises the joints, resulting in chronic pain and movement disorders. The administration of cultured adipose-derived stem cells (ADSCs) into the knee joint cavity improves the clinical symptoms of Knee OA; however, the effect of synovial fluid (SF) filling the joint cavity on the injected ADSCs remains unclear. In this study, we investigated the effect of adding SF from Knee OA patients to cultured ADSCs prepared for therapeutic use in an environment that mimics the joint cavity. An increase in the viability of ADSCs was observed following the addition of SF. Gene expression profiling of SF-treated ADSCs using DNA microarrays revealed changes in several genes involved in cell survival. Of these genes, we focused on FOSL1, which is involved in the therapeutic effect of ADSCs and the survival and proliferation of cancer stem cells. We confirmed the upregulation of FOSL1 mRNA and protein expression using RT-PCR and western blot analysis, respectively. Next, we knocked down FOSL1 in ADSCs using siRNA and observed a decrease in cell viability, indicating the involvement of FOSL1 in the survival of ADSCs. Interestingly, in the knockdown cells, ADSC viability was also decreased by SF exposure. These results suggest that SF enhances cell viability by upregulating FOSL1 expression in ADSCs. For therapy using cultured ADSCs, the therapeutic effect of ADSCs may be further enhanced if an environment more conducive to the upregulation of FOSL1 expression in ADSCs can be established.

## 1. Introduction

### 1.1. Widespread Use of Adipose-Derived Stem Cells (ADSCs)

ADSCs have properties similar to mesenchymal stem cells (MSCs), which are abundant in adipose tissue and may be readily harvested in large quantities in a relatively minimally invasive manner. ADSCs have excellent proliferative potential, and it is possible to culture and prepare the number of cells needed for treatment in a relatively short period of time [1,2]. The therapeutic effects of ADSCs on various diseases are not limited to their ability to differentiate into other cells but also involve the secretion of various cytokines and exosomes that have immunosuppressive, angiogenic, and apoptosis-inhibitory effects [3,4,5]. ADSCs are currently being used clinically for the treatment of Knee OA [6,7,8,9], rheumatoid arthritis [10], pressure urinary incontinence [11], scleroderma [12], skin damage [13], and other disorders. Positive short-term results have been reported for each, indicating that they may be further developed as a safe and effective treatment option for disease treatment.

### 1.2. ADSC Treatment for Osteoarthritis of the Knee

Knee OA is an irreversible degenerative condition that causes chronic pain and movement disorders resulting from the deformation of the bones and the degeneration of the articular cartilage that make up the knee joint. In recent years, cell therapy and regenerative medicine have attracted the attention of patients whose pain cannot be controlled using conservative treatment, such as medication, physical therapy, and intra-articular hyaluronic acid injection, and for whom surgery is not indicated or who do not wish to undergo surgery [6,7,8,9]. Cell therapy and regenerative medicine for Knee OA involves the intra-articular administration of cultured ADSCs. Currently, clinical trials using cultured ADSCs are ongoing, and it has been reported that ADSCs increase the amount of cartilage, based on MRI results [14,15], improve clinical symptoms, such as pain suppression [16], and may delay the progression of Knee OA [17].

In general, when treating Knee OA, frozen ADSCs are used because the cells are readily available in a ready-made state in facilities without the need for cell processing facilities [18,19]. ADSCs are isolated from adipose tissue primarily using enzymatic treatment, expanded culture in a standard medium at 37 °C/5% CO_2,_ and cryopreserved. As shown in Appendix A, frozen ADSCs are thawed and washed before intra-articular injection and then injected into the joint cavity mainly by suspending them in a saline solution [20,21]. In cell therapy of Knee OA using ADSCs, various cell suspensions have been used to administrate ADSCs [14,15,16,20,21,22,23,24,25,26,27,28,29,30,31,32,33,34,35,36,37,38,39,40,41,42,43], such as saline [15,20,21,22,23,24,25,26,27,28,29], PBS [14], and a polyionic solution containing 3.6% human albumin and glucose [30] (Appendix A). Of them, saline is the commonly used solution to resuspend cultured ADSCs in the treatment of Knee OA in clinical trials [15,20,21,22,23,24,25,26,27,28,29]. Saline has also been used as a suspension solution in cell therapy with ADSCs for diseases other than Knee OA [44,45,46,47,48,49,50,51,52].

### 1.3. Availability of ADSCs in the Joint Cavity

In animal studies, it was reported that less than 1% of ADSCs administered systemically survive after 1 week [53]. In addition, ADSCs administered into the knee joint cavity of osteoarthritis model rats disappeared in approximately 2 weeks [54,55,56]. Therefore, to prolong the survival period of ADSCs in the joint cavity, many studies have embarked on the development of spheroid ADSCs [57], improved culture media for ADSCs [58,59], and optimization of the environment prior to administration [18,60]. In cell therapy with ADSCs, the high survival rate and high differentiation potential of ADSCs in the joint are important to achieve therapeutic efficacy [61]. On the other hand, apoptotic ADSCs may exert therapeutic effects on OA owing to their immunomodulatory effects [62]. Thus, multiple mechanisms underlie the therapeutic effects of ADSCs. Manferdini et al. also reported that inflammatory factors and a hypoxic environment in SF are involved in the proliferative and migratory potential of ADSCs [63]. However, there have been few reports on the relationship between the SF that fills the joint cavity where ADSCs are administered and the SF prepared in a realistic clinical manner.

### 1.4. Properties of SF

SF is a viscous fluid that accumulates in the joint cavity. It is a mixture of plasma, hyaluronic acid, and glycoproteins secreted by synovial cells. The electrolytes are similar to that of blood, but the concentration of globulins is low, and fibrinogen is almost nonexistent. SF is involved in the lubrication of the sliding joint surfaces, shock cushioning, and nourishment of cartilage [64]. Patients with Knee OA have decreased levels of hyaluronan in SF as well as increased levels of SF and interleukin (IL)-6, an inflammatory cytokine [65,66,67]. IL-6 in SF has also been reported to activate ADSCs [68]. Therefore, ADSCs administered into the joint cavity are in direct contact with SF, a liquid that is not present in the culture environment or during the administration process.

### 1.5. Effect of SF on ADSCs

When administering ADSCs into the knee joint cavity of patients with osteoarthritis, which is the actual administration environment, it is important to determine the effect of SF filling the joint cavity and its effect on the administered ADSCs. Kiefer et al. reported the effect of adding SF derived from canine osteoarthritis of the knee to canine ADSCs in culture and found that SF may cause cytotoxicity to the ADSCs injected into the joint cavity [69]. Fuku et al. also reported the possibility that SF derived from human Knee OA may have a lethal effect on human ADSC spheroids [70]. However, because these studies involved adding SF to cell culture media that were not used for treatment, the results may not be relevant to the clinical treatment.

In the present study, we compared the effects of SF in vitro with the administration of cultured frozen ADSCs, which is currently used in clinical trials for ADSC therapy. We used saline for the preparation of the ADSC suspension, which is inexpensive, safe to administer, and has been evaluated in several clinical studies [15,20,21,22,23,24,25,26,27,28,29]. In addition, Kiefer et al. and Fuku et al. have examined the change in cell viability caused by the addition of SF, but not at the genetic level. The present study is the first to examine the effects of human SF on human ADSCs at the cellular and gene expression level in clinical specimens. In addition, multiple mechanisms underlie the therapeutic effects of ADSCs, as described in Section 1, Section 2 and Section 3 of the Introduction Section: the expression of FOSL1 in ADSCs has been shown to prevent cartilage damage and suppress inflammation in Knee OA and cancer stem cells involved in the regulation of proliferation and survival [71,72]. Therefore, we focused on cell survival induced by SF exposure of ADSCs and investigated the involvement of FOSL1.

## 2. Materials and Methods

### 2.1. Collection and Evaluation of SF

SF was collected from the knee joints of patients with osteoarthritis [Kellgren–Laurence (K–L) classification grade 2–3] who gave informed consent after obtaining approval from the Kanazawa Medical School Medical Ethics Review Committee (approval number: I583) [73]. Because the administration of drugs, such as NSAIDs (non-steroidal anti-inflammatory drugs), has been reported to alter the immunomodulatory substances in SF [74], we collected SF from untreated patients with Knee OA. After disinfection of the knee region with povidone-iodine, SF was collected by joint aspiration using an 18 G needle using an ultrasound-guided superior lateral approach. The collected SF was centrifuged to remove cellular components and the supernatant was frozen at −80 °C. To determine the concentration of hyaluronic acid and IL-6 in the SF of patients with Knee OA, an ELISA kit (K-1200: Echelon Biosciences Inc., Salt Lake City, UT, USA; D6050: R&D Systems, Minneapolis, MN) was used according to the manufacturer’s protocol. To confirm the presence of blood in SF, the hemoglobin concentration was measured using an automated small six-fraction hematology analyzer XN-450 (Sysmex Corp., Kobe, Japan) according to the manufacturer’s instructions.

### 2.2. Effect of SF on ADSC Viability

Human-derived ADSCs (Lonza K.K., Basel, Switzerland) [75,76] were cultured in a serum-free KBM ADSC-4 medium (Kohjin Bio Co., Ltd., Sakado, Japan) at 37 °C/5% CO_2_. When the purchased ADSCs were thawed and seeded into the dish, the culture was expanded within six passages and used for the experiment. Cultured ADSCs were then detached from cells with 0.25% trypsin-EDTA (FUJIFILM Wako Pure Chemical Corporation, Osaka, Japan), suspended in CELLBANKER^®^ (Takara Bio, Shiga, Japan), and frozen at −80 °C. The cells were then incubated in a KBM ADSC-4 medium and frozen at −80 °C. Just before use, the frozen tubes were placed in a 37 °C water bath and incubated for 5 min to thaw completely before use. Thawed ADSCs were suspended in a serum-free medium, centrifuged, washed with 0.9% saline, and suspended in saline. The ADSCs were then stained with trypan blue to determine the cell count and percent cell viability. A cell viability of 90% or higher was confirmed. The results are shown in Appendix A. Next, 100 μL of saline containing 0%, 20%, 40%, or 80% SF were seeded into each well of a 96-well plate containing ADSCs (1 × 10^5^ cells/mL). After incubation for 30 min, 1 h, 2 h, and 4 h, the CellTiter-Glo 3D Cell Viability Assay (Promega Corp., Madison, WI, USA) was performed based on the manufacturer’s protocol. The GloMax 96 Microplate Luminometer (Promega Corp.) was used to measure the relative light units (RLUs) for each well. The percentage of RLUs in the samples at each time point was calculated by subtracting the RLU of each solution without ADSCs as a blank and dividing by the RLU at time 0.

### 2.3. Morphological Cell Evaluation

After thawing, the cell morphology of the cultured frozen ADSCs with SF (20%, 40%, 80%) and without (30 min, 1 h, 2 h, 4 h) was photographed using a phase contrast microscope ECLIPSE Ts2 (Nikon, Tokyo, Japan) with a 20× objective lens magnification at each time point.

### 2.4. Transcriptome Analysis Using DNA Microarrays

RNA extraction was performed using the RNeasy Mini Kit (Qiagen NV, Venlo, Netherlands) and total RNA was isolated according to the manufacturer’s instructions. Sample concentrations were measured with a NanoDrop Light (Thermo Fisher Scientific Inc., Waltham, MA, USA). RIN (RNA integrity number) values of RNA were determined using the Agilent RNA 6000 Nano Kit (Agilent Technologies Inc., Santa Clara, CA, USA) and Bioanalyzer RNA2100 (Agilent Technologies Inc.) All samples were confirmed to have a RIN >9. Complementary DNA (cDNA) was synthesized using the GeneChip™ WT Amplification Kit (Affymetrix, Thermo Fisher Scientific, Cleveland, OH, USA). Synthesized cDNA was hybridized to the GeneChip™ Human Gene 2.0 ST Array (Affymetrix) using the GeneChip™ Hybridization Oven 645 (Affymetrix), Station 450 (Affymetrix), and scanned using a GeneChip™ Scanner 3000 7G (Affymetrix). Signal intensities were quantified with console software 14.9 (Agilent) and analyzed with GeneSpring ver. 14.9.1 (Agilent) and Ingenuity Pathway Analysis (IPA, Qiagen).

### 2.5. RT-PCR (Reverse Transcription Polymerase Chain Reaction)

cDNA was synthesized using the SuperScript III First-Strand Synthesis System (Thermo Fisher Scientific). The cDNA was subjected to PCR using TaKaRa Ex Taq^®^ (Takara Bio) with PCR primers for β-ACTIN and FOSL1 (β-ACTIN: Forward 5′-GAGCACAGAGCCTCGCCTTT-3′ and Reverse 5′-AGAGGCGTACAGGGATAGCA-3′, product size 494; FOSL1: Forward 5′-AGAACCCAGCAGCCGTGTA-3′ and Reverse 5′-ATGGTGTTGATGCTTGGCAC-3′, product size 175). PCR was performed using a thermal cycler (WAKO WK-0232) (FUJIFILM Wako, Osaka, Japan) with 33 cycles of 10 s at 98 °C, 30 s at 57 °C, and 60 s at 72 °C. PCR products were separated using agarose gel electrophoresis and analyzed using Fusion FX7 Special VILBER LOURMAT (Vilber, Collégien, France). Each band was evaluated quantitatively using ImageJ software [77] and each measurement was calibrated with reference β-ACTIN.

### 2.6. Western Blot Analysis

Proteins were extracted from cells using RIPA buffer (FUJIFILM Wako Pure Chemical Corporation) containing cOmplete™ protease inhibitor cocktail (Roche, Mannheim, Germany). Cell lysates were subjected to SDS treatment with Sample Buffer Solution (FUJIFILM Wako Pure Chemical Corporation) containing 2-mercaptoethanol (Sigma, Darmstadt, Germany). SDS-PAGE was carried out using a 10% separating gel and a 5% stacking gel in 1 × Tris/Glycine/SDS running buffer (Bio-Rad Laboratories, Inc., Hercules, CA, USA). The proteins were transferred to PVDF Immobilon-P^®^ membranes (Sigma) using a blotting apparatus (ATTO, Tokyo, Japan) in 1 × Tris/Glycine buffer (Bio-Rad Laboratories, Inc.). PVDF membranes were blocked with 5% skim milk/TBST or 3% Bovine serum albumin/TBST for 60 min at room temperature and incubated with anti-human β-Actin antibody (1:10,000; A5441; Sigma) or anti-human FOSL1 antibody (1:2000; 5281S; Cell Signaling Technology, Inc., Danvers, MA, USA) diluted in 3% BSA/TBST at 4 °C overnight. After washing, horseradish peroxidase (HRP)-conjugated anti-mouse IgG antibody (1:2000; P0447, Agilent Technologies Inc.) or HRP-conjugated anti-rabbit IgG antibody (1:2000; P0448; Agilent Technologies Inc.) diluted in 3% BSA/TBST was added and incubated for 60 min at room temperature. Bound antibodies were detected with chemiluminescence using Amersham™ ECL Select™ (Cytiva, Tokyo, Japan) and Fusion FX7 Special VILBER LOURMAT.

### 2.7. Knockdown of FOSL1 Gene by Small Interfering RNA (siRNA)

ADSCs (Lonza K.K., Basel, Switzerland) cultured in ADSC-1 (Kohjin Bio Co., Ltd.) were seeded into 24-well plates at 5 × 10⁴ cells/well and incubated with Opti-MEM^®^ (Thermo Fisher Scientific Inc.) and Lipofectamine RNAiMax^®^ (Thermo Fisher Scientific Inc.), which were used according to the manufacturer’s instructions. siFOSL1(1) (s15583; Thermo Fisher Scientific Inc.), siFOSL1(2) (s15585; Thermo Fisher Scientific Inc.) or Negative Control siRNA (siControl) (4390843; Thermo Fisher Scientific Inc.) were transfected at a final concentration of 10 nM. After 24 h, the culture medium was removed and washed twice with cold PBS, and then the protein extracts were prepared as described in Section 2.6. The knockdown of protein expression was confirmed 24 h after transfection using western blot analysis. Each band was quantified using ImageJ software [77] to evaluate knockdown efficiency. The RLUs after a 24 h transfection was also measured as in Section 2.2. The percent RLUs in the samples at each time point was calculated by subtracting the RLU of each ADSC-free solution as a blank and dividing by the RLU at 0 h.

### 2.8. Comparison of the Viability of FOSL1-Knockdown ADSCs and the siControl Group in SF

ADSCs transfected with siRNA using the same method as described in Section 2.7 were frozen as described in Section 2.2. The ADSCs were thawed in the same manner and seeded into 96-well plates at 1 × 10^5^ cells/mL in 100 μL of 0.9% saline for injection with 0% and 20% SF. After incubation for 2 and 4 h, the RLUs were measured as described in Section 2.2. The percent RLUs in the samples from each time point were calculated by subtracting the RLU of each solution without the respective ADSCs as a blank and dividing by the RLU at 0 h.

### 2.9. Statistical Analysis

Differences in survival means were determined using a two-way measure repeated analysis of variance and Tukey’s multiple comparisons test. Gene and protein expressions were analyzed using a Student’s *t*-test. Gene ontology analysis of the ADSCs treated with SF was analyzed using Fisher’s exact probability test. Each data plot is expressed as the mean ± S.D. Spearman’s rank correlation coefficient was used to test the correlation between BMI and IL-6 concentration or cell count and luminescence. All statistical analyses were performed using GraphPad Prism software (version 9.4.1; GraphPad Software Inc., San Francisco, CA, USA) or R (4.2.0).

## 3. Results

### 3.1. SF Properties

At first, we evaluated the properties of the SF collected in our study from untreated Knee OA patient knee joints diagnosed as K–L classification grade 2–3. Age, sex, weight, height, K-L classification, hemoglobin contamination, Body mass index (BMI), hyaluronic acid concentration, and IL-6 concentration of each specimen are shown in Table 1. All SF was clear yellow with medium-to-high viscosity and were collected from three males and three females. The K–L classification was grade 2 in four samples and grade 3 in two. The mean hyaluronic acid concentration was 88.8 ± 10.7 μg/mL and all six samples exhibited a decrease in hyaluronic acid concentration compared with that of normal human SF. No hemoglobin contamination was detected in the samples, and the effect of blood contamination was excluded. These results were consistent with other studies involving SF of Knee OA [65]. It has been reported that SF in Knee OA patients with high BMI has increased levels of inflammatory cytokine IL-6, which increases the risk of OA progression [78,79]. Furthermore, IL-6 is an important mediator of cartilage destruction in OA [80]. Therefore, the correlation between IL-6 concentration in SF and BMI was evaluated. All SF samples were also found to contain IL-6 (Table 1). As presented in Table 1, three of the six patients had a BMI of ≥25 kg/m^2^. However, the concentration of IL-6 in SF did not differ significantly between patients with BMI ≥ 25 kg/m^2^ and BMI < 25 kg/m^2,^ and no correlation was found between BMI and the concentration of IL-6 in SF (Appendix A).

### 3.2. Evaluation of ADSC Morphology Incubated with Various Concentrations of SF over Time

The administration of cultured frozen ADSCs, which is the same condition as clinical trials of ADSC therapy, typically involves injecting ADSCs suspended in saline into the joint cavity, and the ADSCs administered into the joint cavity are gradually exposed to SF. Therefore, we evaluated the effect of exposure to SF on the cell morphology of ADSCs. The ADSCs were suspended in saline solution and incubated with SF at concentrations of 0%, 20%, 40%, and 80% at 37 °C for various times. The cell aggregation was observed only in SF (+) conditions as indicated by the arrows in Figure 1.

### 3.3. Viability of Cultured ADSCs in Response to SF Addition

Next, the effect of SF on the viability of ADSCs was examined using SF samples from six cases. As shown in Figure 1, the ADSCs exposed to SF exhibited cell aggregation. Hence, it was necessary to use a system in which aggregated cells could be analyzed for cell viability, rather than flow cytometry, which requires a single-cell suspension. Therefore, the CellTiter-Glo 3D Cell Viability Assay (Promega), which measures cell viability even if the cell population has a three-dimensional structure, was used to analyze the percentage of viable cells [70]. First, each concentration of SF was verified not to interfere with the assay (Appendix A).

SF concentrations of 0%, 20%, 40%, and 80% were added to ADSCs suspended in saline solution, and the percentage of viable cells (%RLU) was compared after incubation at 37 °C for various times (Figure 2). The SF-untreated group showed a decrease in %RLU over time (*p* < 0.01 at 2 h and *p* < 0.001 at 4 h of incubation). At 30 min, the %RLU was significantly lower in the 80% SF-treated group compared with that in the untreated group (*p* < 0.05). After a 1 h incubation, the %RLU was significantly higher in the group treated with 40% SF (*p* < 0.01) compared with the untreated group. At 2 h, the %RLU was significantly higher in the group treated with 20% SF (*p* < 0.001) and 40% SF (*p* < 0.0001) compared with the untreated group. The %RLU was significantly higher at 4 h compared with the untreated group. On the other hand, the ADSC group suspended in serum-free growth medium exhibited a significant increase in viable cell rate at 4 h compared with the SF-added group. These results indicate that SF treatment improves the viability of ADSCs during preparation for treatment.

### 3.4. Transcriptome Analysis of Changes in Gene Expression following SF Treatment

To explore candidate genes that are associated with SF-induced changes in the viability of ADSCs, total RNA was extracted from ADSC and subject to a microarray analysis. The samples included ADSCs from the 20% SF-treated and untreated groups at 2 h using three SF samples (patient no. #1, #2, and #4). The number of genes, in which expression was increased or decreased in the three samples were extracted at *p* < 0.05 and |log₂ fold change (FC)| > 1, included 27 of 62 genes that were upregulated following SF exposure and 13 of 62 genes that were down-regulated (Figure 3, Appendix A). Furthermore, Gene Ontology (GO) analysis using IPA was performed on the differentially expressed genes, and the results indicated that genes involved in cell death and cell survival were altered (Table 2).

### 3.5. Gene Expression Analysis Using RT-PCR

As shown in Figure 3, the microarray results revealed changes in the expression of many genes. Among them, we focused on FOSL1 in ADSCs because its expression has been shown to effectively protect against cartilage damage in Knee OA and because it is involved in the regulation of cell proliferation and survival in cancer stem cells [71,72]. Semiquantitative RT-PCR was used to evaluate whether FOSL1 mRNA expression was variable in ADCSs suspended in saline with SF. SFs were collected from each of the six patients with Knee OA. The results indicated a significant increase in FOSL1 expression (*p* < 0.01), which was consistent with the microarray results (Figure 4b).

### 3.6. FOSL1 Protein Expression Using Western Blot Analysis

To further validate the microarray results, we evaluated the protein expression of FOSL1 in ADSCs. The increased expression of the FOSL1 was also measured at the protein level in all six samples representing the SF-treated group. All showed elevated expression of FOSL1 (Figure 5a), and quantitation of each band revealed that FOSL1 was significantly upregulated at the protein level with *p* < 0.0001 (Figure 5b). As shown in Figure 4 and Figure 5, increased FOSL1 expression was validated using RT-PCR and western blot analysis, which supports the microarray results (Figure 4 and Figure 5).

### 3.7. Effect of FOSL1 Knockdown on Viable Cell Rate of ADSCs

The RT-PCR and western blot analysis confirmed that FOSL1 expression was increased in the SF-treated group. Therefore, to analyze the association of FOSL1 with cell viability, we knocked down FOSL1 by siRNA and evaluated its effect on the viability of cultured ADSCs. The protein expression of FOSL1 decreased after 24 h in ADSCs in which FOSL1 was knocked down using siRNA of two different sequences (Figure 6a). Furthermore, quantitation revealed that a significant decrease in protein expression of FOSL1 occurred in the siFOSL1 group compared with the siControl group (Figure 6b). When the viability of ADSCs in the FOSL1-knockdown group was compared to the siControl group at 24 h post-transfection, a significant decrease in %RLU was observed (*p* < 0.0001, *p* < 0.01) (Figure 6c). These results indicate that FOSL1 expression contributes to the maintenance of ADSC viability.

### 3.8. FOSL1 Is Involved in the Viability of ADSCs Treated with SF

To further confirm whether FOSL1 is involved in the increased survival of ADSCs in the SF-treated group, we compared the survival rates of FOSL1-knockdown ADSCs with two different siRNA against FOSL1 ((1) and (2)) and the siControl group in SF-added saline (Figure 7). The % RLUs in the siControl group without SF was significantly lower compared with that in the siControl group at 0 h of incubation, which was similar to that shown in Figure 2. Furthermore, the siFOSL1 SF-treated group exhibited a significantly lower %RLU compared with the siControl SF-treated group at 2 h and 4 h. These results indicate that FOSL1 is involved in the SF-induced increase in cell viability.

## 4. Discussion

### 4.1. SF Is Involved in the Maintenance of ADSC Viability

In this experiment, saline solution, which has been reported to be used in many cultured ADSC treatments for Knee OA, was used for analysis. Because saline does not contain any other additives, we believed saline would enable a clear evaluation of the effect of SF on ADSCs. ADSCs suspended in saline alone showed significantly lower viability at 2 and 4 h. A previous study also reported that ADSCs suspended in saline had a viability of approximately 60% at 2 h, which is consistent with the results of the present study [81]. As presented in Figure 2, the average cell viability of frozen ADSCs suspended in saline was 24.6% at 4 h.

The viability of ADSCs was significantly increased in the SF-treated groups compared with the untreated group at 20%, 40%, and 80% SF (Figure 2). Even under the poor environment of saline, the mixing of ADSCs with SF was found to improve cell viability. This indicates that SF promotes cell survival compared with saline, and the presence of basic nutrients, such as glucose and albumin in SF, which are components of human plasma, may be responsible for the increased viability of ADSCs. The cell survival effect of SF was confirmed to be sufficient for up to 4 h, so we evaluated a shorter time period.

Although existing reports have shown SF to be cytotoxic [69,70], these reports were based on a comparison of culture media. ADSCs suspended in a serum-free medium as a positive control in this study exhibited the highest cell viability over time (Figure 2). Therefore, rather than SF being cytotoxic, it may be assumed that SF is not superior to a culture medium properly conditioned for cell culture. In ADSCs suspended in saline, which is a clinically relevant protocol used in this study, SF was effective at increasing cell viability. When ADSCs are injected into the joint cavity, there is a protocol in which the joint cavity is washed to remove SF, and then ADSCs suspended in saline are administered [82]. In this case, the absence of SF in the joint cavity may result in lower ADSC survival.

In previous studies involving ADSC treatment of Knee OA, ADSC was administered in 3~5 mL of saline [15,20,21,22,23,24,25,26,27,28,29]; however, the amount of SF in the normal knee joints was 6.7 ± 2.3 mL [66] and that of osteoarthritis patients was 18.47 ± 4.67 mL [83]. The SF in patients with Knee OA is increased compared with that in healthy subjects. Therefore, the actual environment in the knee joint cavity, in which ADSCs are administered in patients with osteoarthritis, is considered to have an SF concentration of approximately 80%. The results of this experiment suggest that the amount of SF at the time of administration is important for the maintenance of ADSC viability for successful therapy with cultured ADSCs because 80% SF is associated with a temporary decrease in viability. However, further studies are needed to determine the most optimal SF concentration required for long-term cell survival of ADSCs in the joint cavity and its effect on overall efficacy.

### 4.2. FOSL1 Is Involved in SF-Induced Increase in ADSC Viability

Comprehensive gene expression analysis using DNA microarrays revealed that genes related to cell proliferation, migration, and survival varied in expression among the upregulated genes in the three SF-treated groups (Figure 3). One of these upregulated genes, FOSL1, encodes FRA-1, which forms a subunit of the transcription factor AP1 and plays an important role in the differentiation of osteoblasts, chondrocytes, and adipose progenitor cells in response to environmental stress and tumor formation [84,85,86,87,88]. Schwabe et al. reported that ADSCs extracted from adipose tissue of transgenic mice systemically overexpressing FOSL1 and administered to the knee joints of osteoarthritis mice showed preventative effects on cartilage damage and suppression of inflammation [72]. This suggests that FOSL1 may enhance the therapeutic effect in ADSCs. Therefore, we examined whether FOSL1 is also involved in cell survival in ADSCs.

Increased FOSL1 expression was validated using RT-PCR and western blot analysis, which supported the microarray results (Figure 4 and Figure 5). Next, siRNA knockdown of FOSL1 in ADSCs resulted in a significant decrease in cell viability at 24 h (Figure 6). It has been reported that FOSL1 knockdown in cancer cells reduces their viability [89,90], and our findings represent the first time that FOSL1 was shown to be involved in ADSC survival.

To directly verify the relationship between increased ADSC viability in SF and FOSL1, we compared the viability of FOSL1-knockdown ADSCs in SF with that of the siControl group after 2 and 4 h of exposure (Figure 7). The FOSL1-knockdown ADSCs in SF exhibited significantly lower viability compared with the siControl group at both time points. Thus, SF increased the expression of FOSL1 in ADSCs and was involved in the increase in cell viability. However, the viability of FOSL1-knockdown ADSCs with SF was higher compared with that of the siControl group without SF, suggesting that FOSL1 alone may not be responsible for the increased viability of ADSCs in SF.

### 4.3. Involvement of FOSL1 in the Immunosuppressive Effects of ADSCs

Schwade et al. reported a suppressive effect on inflammation by significantly suppressing IL-6 secretion in ADSCs extracted from adipose tissue of transgenic mice overexpressing FOSL1 [72]. In addition, increased concentration of IL-6 in the joint cavity promotes cartilage destruction in Knee OA [80]. Therefore, the increase of FOSL1 in ADSCs using SF treatment may suppress osteoarthritis progression and reduce pain by inhibiting cartilage damage through the suppression of inflammatory cytokine production by ADSCs. However, further studies on the secretory components of ADSCs induced using SF exposure are needed.

### 4.4. Relationship between SF Components and Elevated FOSL1 Expression in ADSCs

Considering the factors that upregulate FOSL1 expression in SF, FOSL1 is activated via the RAS and PI3K, ERK1/2 and STAT3 signaling pathways by the induction of cytokine expression, such as platelet-derived growth factor (PDGF), tumor necrosis factor (TNF), and IL-6 [86,91].

In fact, a comprehensive analysis of gene expression in TNFα-stimulated ADSCs indicated that FOSL1 was upregulated [92]. Furthermore, Nees et al. reported that SF contains TNFα [67]. Therefore, the present results suggest that TNFα in SF may increase the expression of FOSL1 and contribute to the increase in cell viability.

In addition, it has been reported that SF contains growth factors (EGF: epidermal growth factor, PDGF) that are involved in maintaining ADSC viability [93]. These factors may be involved in the maintenance of viability by increasing FOSL1 expression. Interestingly, the results of IPA [70,94] demonstrated that 13 of 40 genes commonly altered by SF treatment in each patient were involved in PDGF-BB (Appendix A). The role of these factors will be examined in the future.

SF in patients with Knee OA also contains various bioactive substances as well as inflammatory cytokines. In joint diseases such as Knee OA, proteolytic enzyme matrix metalloproteinases (MMPs) are thought to be the major mediators of cartilage destruction, and SFs contain various MMPs [95,96]. The MMP expression in MSCs regulates survival, and their proliferation has been reported [97]. Further studies are needed to determine whether exogenous MMPs in SF are involved in the cell survival of ADSCs.

Previous studies have reported increased IL-6 levels in the SF of obese patients with OA [78,79]. In the present study, no difference was observed in the levels of IL-6 in SF between obese patients with BMI ≥ 25 and normal-weight patients (Appendix A). Whether the presence or absence of obesity alters the properties of SF and the expression of FOSL1 requires further investigation.

Differences in the amounts of cytokines and SF in patients with OA depending on the joint site have also been reported [98,99]. Further studies are warranted to determine the effect of SF in OA on the expression of FOSL1 in ADSCs based on the presence of SF in different joint sites.

## 5. Conclusions

This study revealed that ADSC therapy for Knee OA may increase FOSL1 expression by exposure to SF in the joint cavity, which contributes to enhanced cell viability. Based on these findings, targeting FOSL1 in the treatment of Knee OA using cultured frozen ADSCs is expected to be useful in the development of cell preparations to improve long-term cell viability in the joint cavity. Furthermore, for the treatment of Knee OA using cultured ADSCs, it is necessary to develop a treatment method to further enhance the therapeutic effect of ADSCs by increasing the expression of FOSL1 by optimizing ADSC culture conditions.

## 6. Patents

This section is not mandatory but may be added if there are patents resulting from the work reported in this manuscript.

## Figures and Tables

**Figure 1 cells-12-00330-f001:**
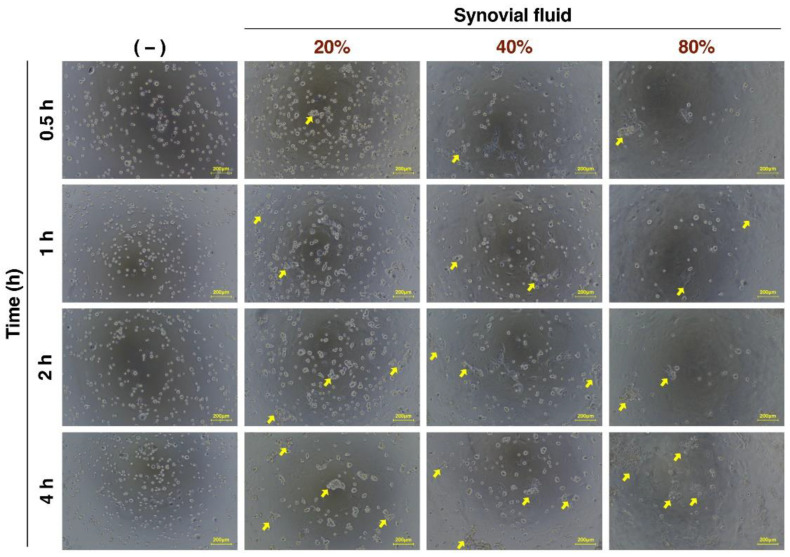
Microscopic images of SF-treated ADSCs over time. Images of cells observed using phase contrast microscopy. Yellow arrow represents aggregated cells. Yellow bars represent 200 μm.

**Figure 2 cells-12-00330-f002:**
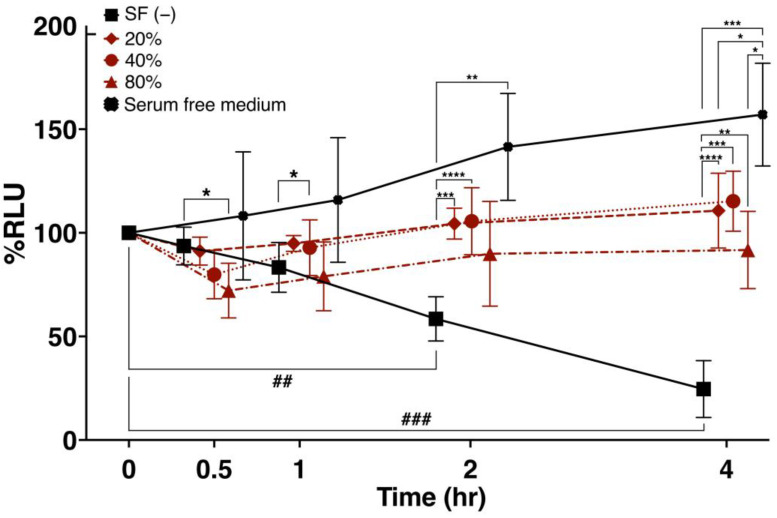
Viability of cultured ADSCs in response to SF. The percent change in ADSC viability following treatment with 0%, 20%, 40%, and 80% SF for various times was evaluated for six SF samples representing Knee OA. SF (−) refers to ADSCs suspended in 100 μL of saline. SF (+) refers to ADSCs in saline containing 20%, 40%, and 80% SF. SF samples obtained from six separate Knee OA patients were used. * *p* < 0.05, ** *p* < 0.01, *** *p* < 0.001, **** *p* < 0.0001, ^##^ *p* < 0.01 and ^###^ *p* < 0.001 are shown. Data are expressed as the mean ± S.D.

**Figure 3 cells-12-00330-f003:**
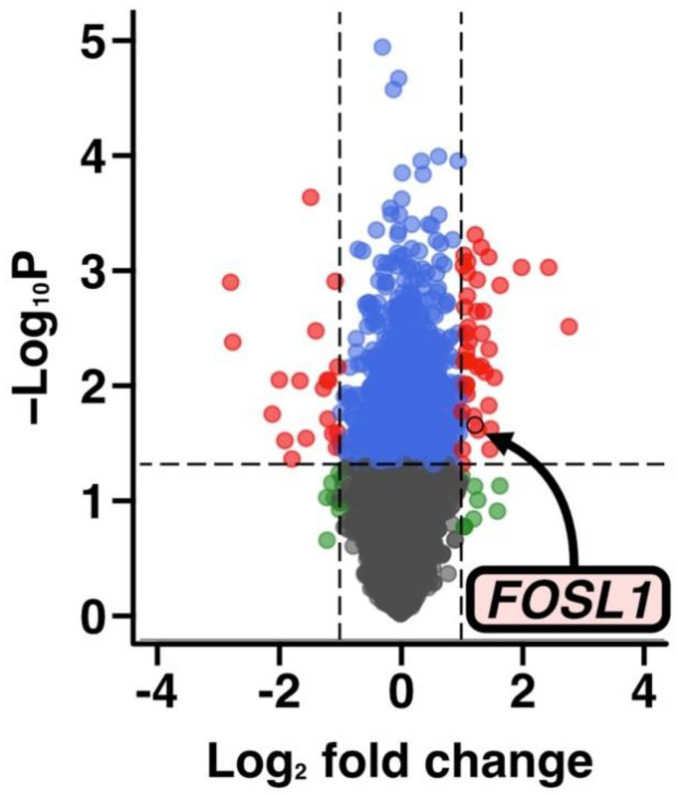
Volcano plot of the DNA microarray results. The volcano plot was created using the microarray results using SF samples from three patients (patient No. #1, #2, and #4). The arrow indicates the FOSL1 gene.

**Figure 4 cells-12-00330-f004:**
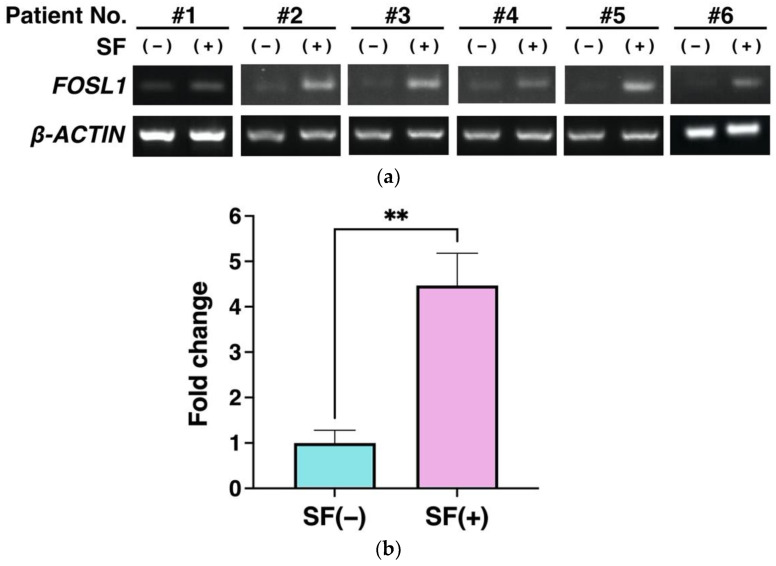
Validation of FOSL1 expression using RT-PCR. (**a**) Agarose gel electrophoresis of the PCR products. (**b**) Bar graph showing the results of quantification of six samples (**a**). Gene expression was normalized to β-ACTIN expression. Data are expressed as the mean ± S.D. ** *p* < 0.01 is shown.

**Figure 5 cells-12-00330-f005:**
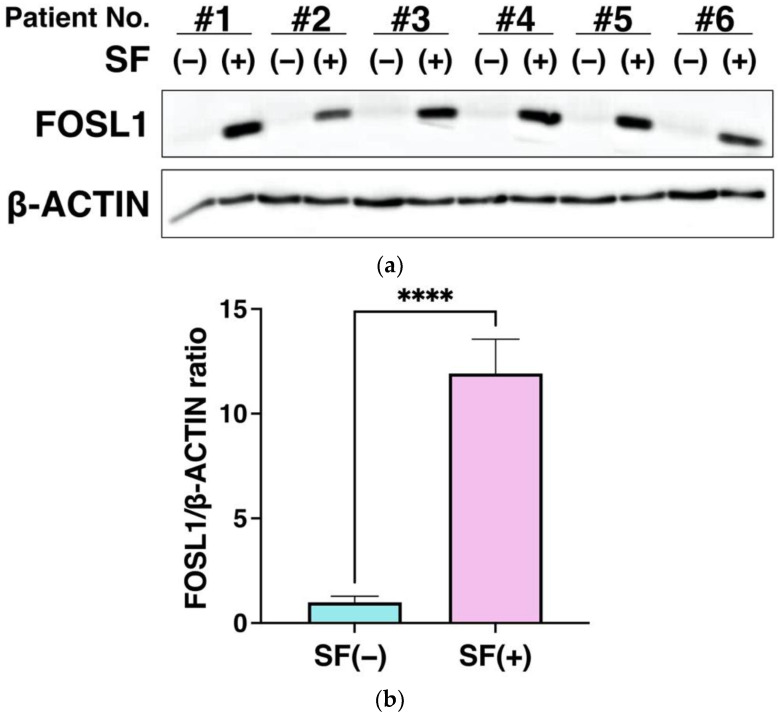
Validation of increased FOSL1 protein expression using western blot analysis. (**a**) Western blot analysis of FOSL1 protein expression. Six SF samples were used to analyze FOSL1 protein levels in the ADSCs. (**b**) FOSL1 expression was quantitated from the bands. Protein expression was normalized to the expression of β-ACTIN and calculated as the relative expression ratio of the group with 20% SF to the untreated group (*n* = 6). Data are expressed as the mean ± S.D. **** *p* < 0.0001 is shown.

**Figure 6 cells-12-00330-f006:**
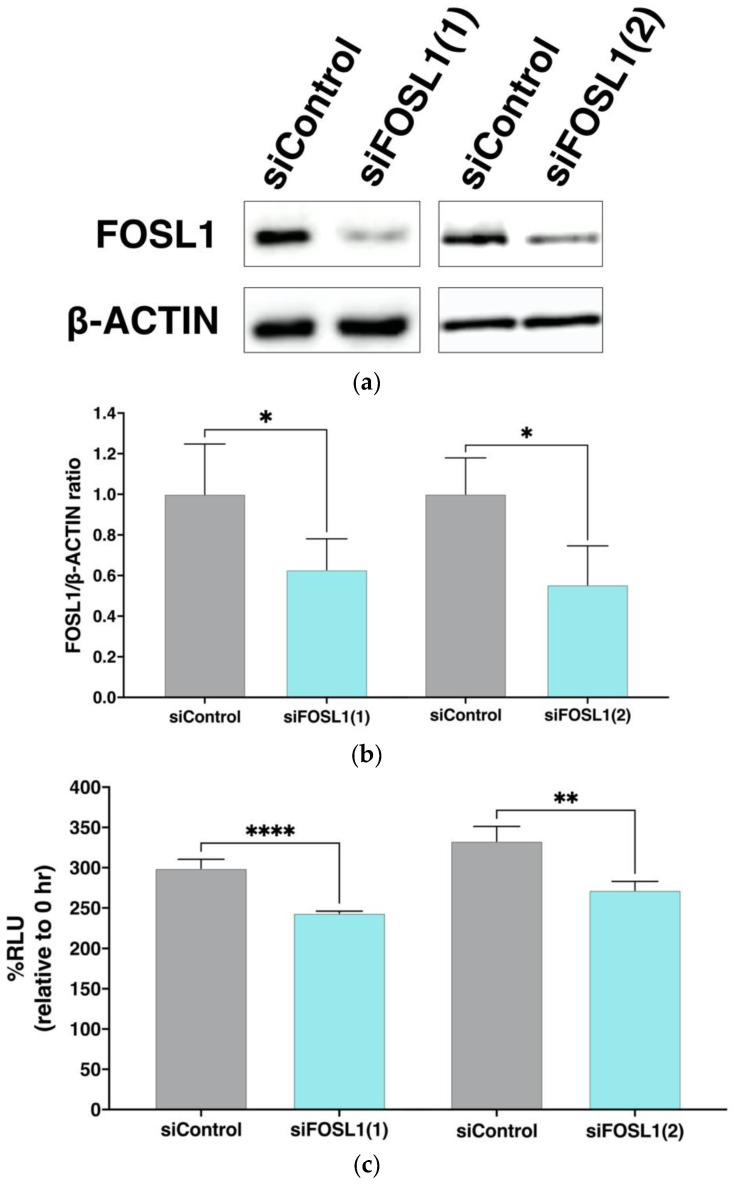
Changes in ADSC viability following siRNA mediated FOSL1 knockdown. (**a**) Expression of the FOSL1 protein using western blot analysis following transfection with two different FOSL1 siRNAs, siFOSL1(1) and siFOSL1(2). (**b**) Six and four independent transfections were performed with siFOSL1(1) and siFOSL1(2), respectively. After 24 h incubation, cell lysates were prepared and analyzed using western blot. FOSL1 expression was normalized to β-ACTIN expression and compared with the siControl group. Data are expressed as the mean ± S.D. * indicates *p* < 0.05. (**c**) Bar graph showing the percentage of viable ADSCs after a 24 h siRNA transfection. The %RLU at each 0 h is expressed as 100% (*n* = 4). Data are expressed as the mean ± S.D. ** *p* < 0.01, and **** *p* < 0.0001 are shown.

**Figure 7 cells-12-00330-f007:**
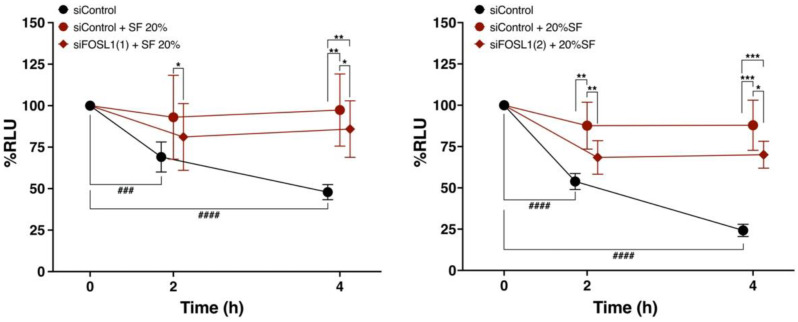
Comparison of percent cell viability between FOSL1-knockdown ADSCs using two different FOSL1 siRNAs and the siControl group following SF treatment. Comparison of changes in percent cell viability over time between FOSL1-knockdown ADSCs and the siControl group following treatment with 20% SF from six Knee OA patients (*n* = 6). SF was diluted in saline. * *p* < 0.05, ** *p* < 0.01, *** *p* < 0.001, ^###^ *p* < 0.001, ^####^ *p* < 0.0001. Data are expressed as mean ± S.D.

**Table 1 cells-12-00330-t001:** Characteristics of patients who donated SF and the properties of SF.

Patient No.	#1	#2	#3	#4	#5	#6
Age	80	81	59	83	73	59
Sex	M	M	F	F	F	M
Weight (kg)	81	60	75	60	46	75
Height (cm)	168	167	161	159	158	160
K–L grading scale	2	2	3	2	3	2
Hyaluronic acid (μg/mL)	74.9	82.3	97.3	82.4	103.5	92.2
BMI (kg/m^2^)	28.7	21.5	28.9	23.7	18.4	29.3
IL-6 (pg/mL)	1462.6	395.3	13,111.7	3866.9	798.9	2808.1

**Table 2 cells-12-00330-t002:** GO analysis of the DNA microarray results.

Molecular and Cellular Functions	*p*-Value Range	#Molecules
Cellular Movement	3.00 × 10^−3^–1.15 × 10^−9^	ABL2, ARRDC3, CCN2, CXCL2, DKK1, DUSP1, ETS1, ETS2, FOS, FOSL1, HBEGF, LINC00312, mir-154, mir-221, mir-29, NFATC2, NFKBIA, SPRY2, DDIT4, USP38, CXCL2
Cellular Development	3.86 × 10^−3^–1.73 × 10^−9^	ABL2, AMOTL2, ARRDC3, CCN2, CXCL2, DAGLB, DDIT4, DKK1, DUSP1, DUSP10, DUSP14, ETS1, ETS2, FOS, FOSL1, HBEGF, HEY1, LINC00312, mir-154, mir-221, mir-29, NFATC2, NOCT, NFKBIA, RGS2, SERTAD1, SPRY2, USP38, VGLL3, ZSWIM6
Cellular Growth and Proliferation	3.86 × 10^−3^–1.73 × 10^−9^	ABL2, AMOTL2, ARRDC3, BNC1, CCN2, CXCL2, DAGLB, DKK1, DDIT4, DUSP1, DUSP10, DUSP14, ETS1, ETS2, FOS, FOSL1, HBEGF, HEY1, LINC00312, mir-154, mir-221, mir-29, NFATC2, NFKBIA, RGS2, SERTAD1, SPRY2, USP38, VGLL3, ZSWIM6
Cell Death and Survival	3.77 × 10^−3^–1.18 × 10^−7^	ABL2, CCN2, CXCL2, DDIT4, DKK1, DUSP1, DUSP10, DUSP14, ETS1, ETS2, FOS, FOSL1, HBEGF, HEY1, mir-154, mir-221, mir-29, NFATC2, NFKBIA, SPRY2, TJP2, USP38
Cell Cycle	2.55 × 10^−3^–7.48 × 10^−7^	CCN2, DDIT4, DKK, DUSP1, ETS1, ETS2, FOS, FOSL1, HBEGF, HEY1, mir-154, mir-221, mir-29, NFATC2, NFKBIA, NOCT, SERTAD1, SPRY2, UBC

The results of a GO analysis of the ADSCs treated with SF. The *p*-values were calculated using Fisher’s exact probability test. The #Molecules column indicates the number of genes associated with the GO term.

## Data Availability

The authors declare that the data supporting the findings of this study are available upon request.

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
