# Peer review of "Synovial Fluid Derived from Human Knee Osteoarthritis Increases the Viability of Human Adipose-Derived Stem Cells through Upregulation of FOSL1"

_cells, 2023, doi:10.3390/cells12020330_

Round 1

Reviewer 1 Report

Overall the paper is sound, but rather limited in scope currently. And there are some points that need clarifying before considering publication

There is some confusion currently about samples and patient numbers both in the text and the figures. There are 6 patients with knee OA from which synovial fluid was collected. And separately there are some ADSCs.

1. In figures 4 and 5 it is not clear what the (-) is for each patient? The (+) represents that patients synovial fluid. But what is the negative? and is this different for each of the 6 patients?

2. In line 306-307. The authors state "We used semi-quantitative RT-PCR to confirm whether there was an increase in the expression of FOSL1 in the six SF samples"  

But surely the authors are measuring FOSL1 in the ADSCs that have been treated with or without patient synovial fluids? At the moment it reads as the authors have measured FOSL1 in the patient synovial fluids and found it to be increased? In which case in what is in increased relative to?

If, I understand correctly the FOSL1 has been measured in the ADSCs.  There are 6 synovial fluids from 6 different patients.  Again here what is the negative control? Is it the same for each patient?

3. In table 1, could the authors include patient BMI. Height and weight has been recorded. But it would be good to include the calculated BMI and interestint to see. It is likely that several will be overwweight or obese. and the authors should comment on this and the likely effect of obese synovial fluid in particular having these effects.

The paper by Mark J Pearson et al should be cited here doi: 10.1038/s41598-017-03759-w.

The paper by Dominika Nanus et al should also be cited showing how obesity makes the snovial tissue more inflamamtory  

https://doi.org/10.1002/art.41158

4. Figure 6. The legend refers to an n=6. The authors should show this in the western blot. Also, could they clarify what the n=6 refers to and was there a second positive siRNA sequence used in the knockdown.

5. I couldnt see the rest of the microarray data? the authors refer to other genes being affected (27 out of 62 genes) by the synovial fluid treatment. what were these other genes? and were any others followed up for analysis?

6. Again, for the microarray experiemnt, why were those particular n=3 patient synovial fluids selected?

7. In the discussion. The authors should discuss how these effects may or may not extrapolate to other patient cohorts. OA is highly heterogeneous, and as mentioned in point 3 above. Obesity makes the synovial fluid more inflammatory. Would such effects be seen in patients of normal weight?  Or in other OA joints such as the hands or hips?

Author Response

We sincerely thank you for your feedback and for reviewing our manuscript (Manuscript ID: ijms-2071866). The comments have helped us significantly improve the paper. 

Reviewer 2 Report

The authors have evaluated the effects of synovial fluid from human osteoarthritis patients on the viability of human adipose-derived stem cells showing that occurs through the upregulation of FOSL1. The main concern of the study is related to the methodology used and the lack of different information. The authors have compared ASC resuspended in saline versus ASC in different concentrations of synovial fluid evidencing that saline was the method mainly use in clinical trial for injecting these cells. This choice do not take in consideration a huge number of other trials that have considered this point and used other solutions to resuspend the cells, like saline+5% AB plasma (Stem Cells Transl Med. 2019 Mar; 8(3): 215–224) or PBS with 2% human albumin ( Knee Surg Sports Traumatol Arthrosc . 2020 Jun;28(6):1989-1999) or PLASMA-LYTE A solution (Arthritis Res Ther. 2016 Dec 20;18(1):301) or 3.6% human albumin and a polyionic solution containing glucose (Stem Cells Transl Med . 2016 Jul;5(7):847-56. doi: 10.5966/sctm.2015-0245)

 The main points are:

1.Introduction. The authors have not reported important studies that have demonstrated that most ASCs disappear 10 days post-intra-articular (IA) injection in a murine model of OA (Toupet, K, PLoS ONE 10:e0114962. doi: 10.1371/journal.pone.0114962, Arthritis Rheum. 65,1786–1794 doi: 10.1002/art.37960)

However, despite death and clearance of administered cells, significant therapeutic effects are observed in response to IA injection of mouse ASCs. Moreover, it has been shown the important role of apoptotic ASC in counteracting the OA disease ( Mancuso P, Frontiers in Bioengineering and Biotechnology  doi: 10.3389/fbioe.2019.00009). Finally, it is known that inflammation and hypoxia affect ASCs in OA milieu (Manferdini C, J Orthop Res. 2020 Feb;38(2):336-347. doi: 10.1002/jor.24446.). All these basic concepts demonstrated by different groups were not considered in the Introduction as well as the role and function of FOSL1 and other markers involved in cell viability

2.It is necessary to insert the experimental scheme since different passages are not clear

3.Why the authors have tested in synovial fluid only hyaluronic acid and not inflammatory factors known to activate ASCs?

4.It is unclear if they have used pooled synovial fluids or not and if the experiments were done with one or more ASCs. Moreover, the use of only saline as control do not take in consideration all the other solutions used to resuspend ASCs for clinical trials versus a solution with FBS as positive control.

5.In Table 1 some data were not reported

6.In Figure 1 they have observed an increase of cell migration due to an increase concentration of synovial fluid, but they have not done a quantitative migration test (i.e. transwell migration test)

7 They performed viability test by CellTiter-Glo 3D cell viability assay, that is based on quantitation of the ATP present, which is a marker for the presence of metabolically active cells. Have the authors correlated the cell number with luminescence (RLU)? Have the authors considered the ATP content in ASCs? Have the authors tested the concentration of ATPases in synovial fluid? It is known that ATPases are present in serum. In the protocol of the assay is indicated that the chemical environment of the luciferase reaction affects the enzymatic rate and thus luminescence intensity and differences in luminescence intensity have been observed using different types of culture media and sera. Have the authors considered if the synovial fluid interferes with the assay?

8. In section 3.4 The authors reported results of transcriptome analysis. They only report GO analysis (as molecular and cellular function and number of molecules) but not reported the molecules identified.

9. Finally, considering all the points raised also the discussion need to be improved.

Author Response

We sincerely thank you for your comments and for reviewing our manuscript (Manuscript ID: ijms-2071866). The comments have helped us significantly improve the paper. 

Reviewer 3 Report

Adipose-derived stem cells (ADSCs) are of interest in treating osteoarthritic joints through intra-articular injection. The process can involve thawing a vial of cryopreserved cells and resuspending them in saline before injection into the synovial space. The final contribution of cells/saline in the synovial fluid will likely range between 20% and 80%. In this study, authors examine the viability and performance of saline-resuspended ADSCs in 20% to 80% synovial fluid from OA donors. In addition, the authors explore the role of FOSL1 in maintaining ADSC viability. The study's main findings are that 100% saline is not tolerated well by ADSCs and that supplementation with 20%-80% synovial fluid increases the survival of ADSCs. A role for FOSL1 in ADSC survival is indicated by knock-down experiments and supported by correlations to FOSL1 protein and mRNA expression in ADSCs.

This is a sound manuscript, well-written, logically organized, and with appropriate experimental design and controls. The overall impact may be minor. There are some minor concerns that would benefit the manuscript if addressed.

1- Freezing of the synovial fluid for experimental use may affect the activity of proteolytic enzymes that would be present in an osteoarthritic knee joint. The authors should discuss this in their evaluation of SF cytotoxicity.

2- The authors provide short time points for analysis, from 1-4 hours. It would help readers if the authors provided a rationale for this earlier in the manuscript.

3- Figure S2 - it would be good to show summary statistics (mean, standard deviation) in addition to each data point.

4- Lines 188-189 - it would be helpful to also describe cOmplete(tm) as a protease inhibitor cocktail.

5- Lines 250-252 - correct grammar, paragraph begins with "And, " and there appears to be missing text

6- Figure 2 - An additional control might include a nutrient-containing substitute for saline. This could provide a solution to the problem identified by the authors (poor viability of freshly thawed ADSCs in saline).

7- Can the authors provide a rationale for conducting most of the experiments (i.e., Figure 7) using only a single 20% SF concentration, as opposed to 20%, 40%, and 80% like they did in the earlier figures?

8- Line 385 - the cytokine concentration in OA synovial fluid is plausible, but was not actually tested in this manuscript.

Author Response

Response: We thank you greatly for your response and for reviewing our manuscript (Manuscript ID: ijms-2071866). The comments have helped us significantly improve the paper. 

Round 2

Reviewer 1 Report

thank you for making the amendments to the manuscript as suggested

Reviewer 2 Report

The authors have answered to all requests